# Too much focus on your health might be bad for your health: Reddit user's communication style predicts their Long COVID likelihood

Ludovica Segneri[1], Nandor Babina[2], Teresa Hammerschmidt[3], Andrea Fronzetti Colladon[1], Peter A. Gloor[4]*

1 Department of Engineering, University of Perugia, Perugia, Italy, 2 Applied Information and Data Science, University of Applied Sciences Lucerne, Lucerne, Switzerland, 3 Department of Business IT, University of Bamberg, Bamberg, Germany, 4 MIT Center for Collective Intelligence, Cambridge, MA, United States of America

* pgloor@mit.edu

**Data Availability Statement:** The dataset containing the numerical computations underlying

## Abstract

Long Covid is a chronic disease that affects more than 65 million people worldwide, characterized by a wide range of persistent symptoms following a Covid-19 infection. Previous studies have investigated potential risk factors contributing to elevated vulnerability to Long Covid. However, research on the social traits associated with affected patients is scarce. This study introduces an innovative methodological approach that allows us to extract valuable insights directly from patients' voices. By analyzing written texts shared on social media platforms, we aim to collect information on the psychological aspects of people who report experiencing Long Covid. In particular, we collect texts of patients they wrote BEFORE they were afflicted with Long Covid. We examined the differences in communication style, sentiment, language complexity, and psychological factors of natural language use among the profiles of 6.107 Reddit users, distinguishing between those who claim they have never contracted Covid -19, those who claim to have had it, and those who claim to have experienced Long Covid symptoms. Our findings reveal that people in the Long Covid group frequently discussed health-related topics before the pandemic, indicating a greater focus on health-related concerns. Furthermore, they exhibited a more limited network of connections, lower linguistic complexity, and a greater propensity to employ emotionally charged expressions than the other groups. Using social media data, we can provide a unique opportunity to explore potential risk factors associated with Long Covid, starting from the patient's perspective.

## 1. Introduction

Post-acute sequelae of Sars-CoV-2 infection (PACS), widely known as Long Covid–is a chronic disease referring to individuals struggling with long-term symptoms after a COVID-19 infection [1], which affects "at least 65 million individuals worldwide. . ., with cases increasing daily" [2]. COVID-19 is a viral-onset illness spread through the coronavirus (SARS-CoV-

the statistical models presented in the paper is now available at: 10.6084/m9.figshare.25251316

**Funding:** The author(s) received no specific funding for this work.

**Competing interests:** The authors have declared that no competing interests exist.

2); approximately 10% of those affected develop Long Covid with more than 200 possible symptoms that affect the organism [2]. The most experienced symptoms of affected people are, for example, fatigue, chest pain, headache, abnormal breathing, anxiety, depression, and decreased physical performance [3, 4]. Furthermore, patients stated that Long Covid affects their mental health due to the constantly changing course of the illness and its symptoms [5]. Thus, many affected people cannot return to work or maintain a social life [6], leading to permanent employment loss and increased hospitalization worldwide [7, 8]

Long Covid significantly impacts public health and people's lives, which "is placing an increased burden on individuals and societies" [4]. Therefore, researchers and healthcare professionals put much effort into investigating links to other post-viral illnesses to increase knowledge on risk factors for prevention and medical treatments for recovery strategies [2, 4]. Given that there are not yet validated effective Long Covid treatments for recovery [2], prevention strategies that focus on reducing possible risk factors become relevant for contributing to public health [8].

Previous research investigated possible risk factors determining a higher probability of individuals falling ill with Long Covid. Such Long Covid traits are: sociodemographics as women, the elderly, and individuals with lower socioeconomic status are at higher risk of developing prolonged symptoms [1]; the severity of Covid -19 disease as hospitalized patients and patients at greater risk [9]; medical comorbidities as patients with obesity, asthma, and diabetes are at greater risk [1, 2, 10] and the vaccination status as the vaccinated population is at lower risk [1, 2, 10]. Although these studies provided valuable contributions to identifying risk factors that predict Long COVID, 'fully understanding complex and multifaceted health conditions requires approaches that capture and amplify the voices of those affected' [8]. Therefore, more prospective research from the patient's perspective considering social traits, such as social exchange and Covid long haulers, is required to increase our knowledge of possible risk factors that distinguish between people being affected by long-term Covid symptoms and those not after a SARS-CoV-2 infection [8].

With this work, we propose a methodological approach that allows us to capture insights directly from patients' voices by extracting information about possible risk factors from written texts posted on social media platforms.

Following Thompson et al. [11], the uncertainty of Covid long haulers due to missing medical cues leads to more people using social media as a crowdsourced medicine. Therefore, social media data can help "to better understand Long Covid from the perspective of patients" [12]. For example, affected people reported long-term loneliness [13], changes in social support [14], and antisocial behavior [3, 15, 16] that may be illustrated through changes in the communication style and the network structure of social media users. As Sarker and Ge [12] determined, before Long Covid, people communicated much on health-related issues on Reddit, while after being infected, participation was reduced, resulting in a decreasing network structure and communication. Social media platforms like Reddit provide a reasonable basis for Long Covid studies [12, 17]. This lets us assume that social media data might be appropriate to derive possible social traits of Long COVID by exploring the communication style and the social network structure within user profiles. This also refers to the calls of Nittas et al. [8] and Burton et al. [18] to use long-term investigations to better understand Long Covid from the patient's perspective.

Considering this scenario, our objective is to investigate possible social traits of COVID-19 patients that influence the possibility of a Long Covid disease. With the help of text analysis and Social Network Analysis (SNA), we analyzed 6.107 Reddit users' profiles, comprising their posts or comments, by comparing the differences within the communication style and network structure of people who claim that they have never contracted Covid -19 (No Covid

Group, NC), those who maintain that they have had it (Covid Group, C), and those who claim that they have experienced Long Covid symptoms. The aim is to identify social factors that classify covid long haulers (Long Covid Groups, LC). An innovative feature of our analysis is that we identified the Covid status of a large random sample of Reddit users, and then collected their texts from the time before Covid existed.

To identify changes in the communication style and network structure that differentiate Reddit users belonging to the NC, C, and LC groups, we used textual characteristics of the user profiles and SNA metrics based on a constructed social network graph (representing our independent social traits variables). We used a pairwise comparison to investigate differences regarding our three groups (NC, C, and LC as dependent variables). Furthermore, we performed logistic regression models to determine the impact of different social traits variables on the probability of belonging to LC group.

The paper is organized as follows. In Section 2, we discuss the theoretical background and the development of hypotheses. In Section 3, we illustrate the method, describing the data set, data collection, and the construction of independent and dependent variables. In Section 4, we present the results and discuss them in Section 5. Finally, in Section 6, we show the main contributions of the work, highlighting its limitations and possible future developments.

## 2. Theoretical background and hypotheses development

Following our aim to investigate Reddit user profiles' communication style and network structure as possible social traits of Long Covid, we provide hypotheses guiding our analyses based on related work.

We start with our basic assumption that social media data can be used to predict the health of its users. This is based on well-established research. For example, Choudhury and De [19] examined social factors driving social support for the mental health of Reddit users, and Shen et al. [20] investigated early signs of depression among Twitter users. Given that social media platforms provide a rich source of user information based on user-generated content comprising their feelings and thoughts [19, 20], research began to use social media platforms for public health studies [21, 22]. In the context of Covid or Long Covid, there is also initial research using social media data. For instance, Sarker et al. [17] extracted self-reported Covid-19 symptoms of Twitter users; Sarker et al. [12] elaborated Long Covid symptoms of Reddit users. Reddit is especially appropriate for Covid-related studies, as the social media platform with 48 million active monthly users exhibits several Long Covid groups (organized in subreddits, such as r/covidlonghaulers with over 30,000 members) [12]. Therefore, our work builds upon the assumption that Reddit's social media platform provides a reasonable basis for analyzing the social traits of Long Covid.

Looking deeper into possible relations between the communication style and the probability of coming down with Long Covid, we searched for mental health studies demonstrating how anomalies within communication can indicate chronic diseases. In general, while positive emotions can be associated with immunity against specific diseases, negative emotions can be related to an increased risk of developing a disease [20, 23]. For example, depressed people often express more negative emotions when communicating on social media [20]. A positive emotional style can predict resistance to illness after experimental exposure to rhinovirus or influenza A virus, which are other viral-onset illnesses compared to COVID-19 [24]. Given that–similar to depression–Long Covid can be related to chronic diseases and–similar to Influenza A virus–to a viral-onset illness with a significant impact on mental health [1, 2, 12], equal effects can be assumed.Hence, we formulate this first hypothesis to see whether:

H1: *Long Covid users express more negative emotions compared to Covid-19 users (not having long-term symptoms).*

Following Zhao and Zhou [25], COVID-related content in social media can be associated with users having worse mental health. Following the authors, users searching for disease-related content reported more significant disaster stressors. In comparison, people more concerned with organizing leisure activities and communicating about family and hobby-related content on social media reported higher well-being and less stress [26, 27]. This is consistent with research on linguistic traits of an individual's health (see, for example, Pennebaker and King [28] who determined a causal link between the use of words by individuals and their health or illness). Hence, excessive use of social media for information and communication about health-related topics, such as the coronavirus pandemic, can impact perceived disease symptoms. This is also known as hypochondriacal beliefs affecting disease progression [29–31]. Mahat-Shamir et al. [32] confirmed the mediation effects of hypochondriasis symptoms of social media users in the context of the COVID-19 pandemic. Hence, we assume similar effects for Covid long haulers:

H2: *Long Covid users express more health-related topics on social media than Covid-19 users.*

Concerning further possible social traits of Long Covid within the communication style of social media users, we elaborate on the language and communication style of the posted content. Before the coronavirus pandemic, psychological studies investigated how linguistic complexity can predict emotional stress [33]. Following Karabin et al. [34], in the context of the Covid-19 pandemic, there has been "a steady increase [in linguistic complexity] from the pre-pandemic level throughout the first year of the global lockdown". The results were contrary to the author's assumption that linguistic complexity will decrease with infection, given that a loss of cognitive function is associated with COVID-19. This might be because sick people tend to demonstrate a greater length of postings to provide clear and detailed information regarding their physiological and psychological health status [35, 36]. This demonstrates a priority on communication clarity leading to linguistic complexity in terms of word counts and direct communication but not in terms of using polished language with complex words to express their thoughts, since sick people face greater levels of stress (Suefeld & Rank 1996). Since Covid long haulers are uncertain about how to deal with the varying symptoms during their long-term illness [5] leading to a greater use of social media for medical research [11], affected people might try to overcome this uncertainty by adding value through clear and direct postings explaining their illness resulting in a higher word count of verbose language and a more direct communication style. Hence, we assume:

H3: *Long Covid users use more verbose and direct language than Covid-19 users.*

In addition to having communication style as one possible social traits increasing the probability of coming down with Long Covid, we also believe that social media exposure somewhat predicts Long Covid illness. For example, healthy people's communication activity is higher than that of non-healthy ones [19]. In the context of COVID-19, Gao et al. [37] determined that social media engagement and mental health are frequently interrelated. Although self-promotion through social media can benefit mental health [38], cognitive overload with COVID-19 information or misinformation can also result in cyberchondria as "shar[ing] news without verifying its reliability" [39]. Hence, highly active social media users require healthy skepticism [39]. Therefore, users with high engagement within social media might have a reduced probability of developing Long Covid. However, the study of Nicholls and Yitbarek [40] examined that social media engagement does not necessarily lead to more preventive behavior during

the COVID-19 pandemic. Since lethargic behavior can be one post-Covid syndrome [41], Covid long haulers might demonstrate lower engagement within social media. Thus, we assume the following.

H4: *Long Covid users have lower social media communication activity (posting, commenting) than Covid-19 users.*

Besides the communication activity on social media, how individuals are connected with other users (referring to the network structure) can further affect their health [42]. For example, social support and social connectedness on social media are related to lower levels of depression and anxiety [43]. Thereby, "depressed users tend to build a close network of trusted people to share their mental health issues", whereby "a lower value indicates fewer interactions" [42]. In the context of the COVID-19 pandemic, people experience loneliness by having less contact with friends [44]. This can lead to mental health problems, such as poorer physical performance or more chronic conditions [45]. Hence, 'students who kept all-around contact with friends during the lockdown declined in loneliness, whereas students who had little contact or did not (video) call friends did not' [44]. Additionally, especially Covid long haulers reported long-term loneliness [13], reduced social support [14], and antisocial behavior [15, 16, 46], so we assume:

H5: *Long Covid users have fewer social media connections than Covid-19 users and are less central in the social network.*

In summary, while the first three hypotheses (H1-H3) are related to the communication style of social media users as social traits of Long Covid, the last two hypotheses (H4-H5) are related to the network structure of social media users.

## 3. Methodology

The purpose of this research is to understand how the communication style and network position of Reddit users differ between those who have Covid, Long Covid, and those who have never been infected. In this Section, we present our study design based on the methodology used by Shen et al. [20] in their depression detection research, with slight adjustments based on the proposals of Chancellor & De Choudhury [23] regarding how to improve social media-based public health studies. Fig 1 presents the complete process, detailed in the next paragraphs.

### 3.1. Data collection

To obtain our initial dataset, we used the Pushift API, a community-driven Reddit API. We collected information between 1 January 2018 and 1 May 2022. Subsequently, the collected data were divided into two datasets according to their publication date. The first contains all user posts made before the advent of the pandemic (from January 2018 to 1 January 2020). Consequently, the second includes the posts published after the arrival of the pandemic. This second dataset allowed us to label the pre-pandemic posts into three distinct groups of users: users who declared that they did not develop Covid symptoms (No Covid group); users who claimed to have Covid without developing Long Covid (Covid group); users who declared that they developed Long Covid (Long Covid group).

All the posts included in the analysis are written in English. To identify whether a user belongs to one of the three groups, we followed the approach of Chancellor & De Choudhury [23], illustrated in Fig 2.

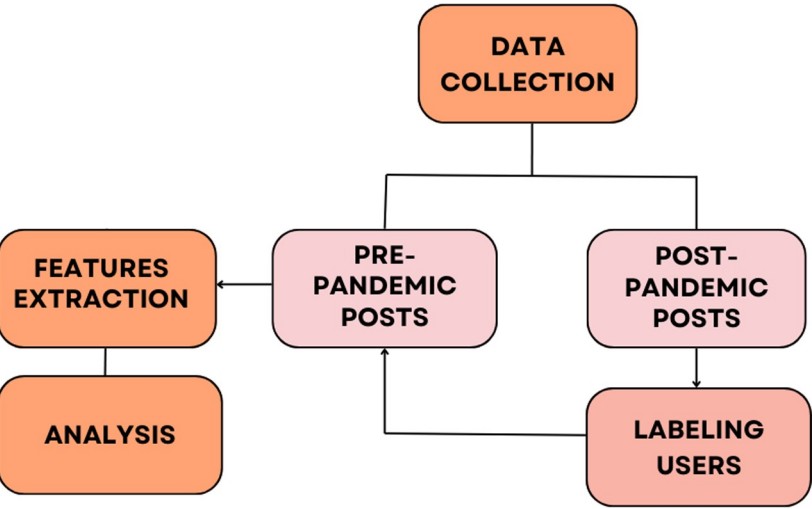

**Fig 1. Research framework.**

In brief, to create the Long COVID group, we identified 2.986 users who posted on the *covidlonghaulers* Reddit forum. Accordingly, we collected all their posts published in this subgroup. The forum has been previously validated by Sarker & Ge [12] for its representation of people with Long Covid symptoms.

The two remaining groups (i.e., Covid and No Covid) are constructed by selecting authors of randomly selected Reddit posts. In the random selection process, all posts within the given timeframe have the same chance of being selected. The random selection is done without replacement. The criteria for categorizing a user as COVID-positive are the active participation in a subforum dedicated to COVID-19-positive patients and a user's explicit declaration of being infected with COVID-19. Userss meeting these criteria are allocated to the Covid group, while those who have never posted in a Covid subreddit and have never explicit stated

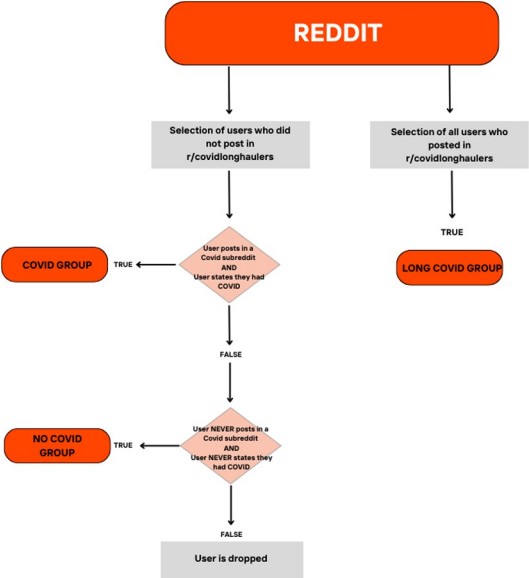

**Fig 2. Construction of the three user groups.**

that they are infected with COVID-19 are assigned to the No-Covid group. In all other cases, the user is dropped. For the first criterion, we listed all subreddits in our data that had the word COVID in it,and identify COVID-19 subreddits dedicated to people infected by the virus. For example, our list includes: r/COVID19positive, r/COVID19, and r/CovidVaccinated sub-reddits. The second criterion is based on a regular expression. This method has been employed in several studies on identifying disease infections on social media, as demonstrated by Chancellor et al. [23]. The aim of this approach is to identify users expressing thoughts regarding their COVID-19 infection. In the, S1 Fig provides several examples of regular expressions used to identify mentions of COVID-19 infection in user posts, while S2 Fig illustrates an example of how these indicators allow us to identify mentions of COVID-19 infection in user posts.

For the validation phase, we manually reviewed 500 posts to ensure the classes were correctly assigned. During the initial phase, we noticed classification errors stemming, for example, from the definition of "being infected". If a user stated "My father was infected" instead of "I was infected" the classification was incorrect. With the aid of manual review, we refined the regular expression until the issues identified during validation were resolved. Once all major issues were addressed and posts from randomly selected users were consistently assigned to the correct categories, we completed the validation process.

This snowball approach has been used successfully by Chancellor & De Choudhury [23] to identify mental health-related subreddits.

## 3.2. Features extraction

To extract the variables related to the SNA and linguistic aspects of the NC, C, and LC groups, we processed the first dataset, i.e., the one containing posts published by users before the pandemic, from January 2018 to January 2020. To respect the privacy of the users in our sample, we first anonymized the users' names to protect their identities and safeguard their health information related to COVID-19 infection. After classifying users into groups and extracting their pre-pandemic posts, we assigned each user a unique ID, ensuring that specific names were not retained in our internal databases. Additionally, we did not make the analyzed texts public. Instead, we present only aggregated measures that describe the characteristics of the three groups, ensuring that the textual content of individual users cannot be traced back to them. Numerical results are publicly available at Fighshare (10.6084/m9.figshare.25251316).

**3.2.1. Network construction and SNA variables.** We constructed a social network graph considering users' interactions in our database. The graph consists of nodes representing users and edges representing comments exchanged between two users. The network is weighted according to the number of comments user $i$ makes to user $j$'s posts. To construct a network, the recipient of a post was set as the original poster triggering the response. When it was not possible to directly identify the recipient of a post, we made the corresponding subreddit a virtual recipient of a post. Using this methodology, we constructed a comprehensive network that includes users and subreddits as nodes, with posts or comments as connecting edges. This network representation allowed us to analyze centrality metrics and gain insights into the prominence of users within the platform.

The social network features were retrieved using Griffin, a web-based social network analysis tool [47]. The variables *messages sent* and *messages received* provide quantitative measurements of the total number of messages sent and received by each user in our study. To further analyze these communication data, we explored the concepts of *in degree* and *out degree* centrality [48]. In degree centrality quantifies the number of incoming arcs directed towards a user $i$, specifically representing comments made by any user in response to user $i$'s posts.

Conversely, out degree centrality captures the number of outgoing arcs originating from a user *i*. In addition, we calculated *betweenness centrality*, which measures how often a node occurs on all shortest paths between two nodes [49].

To account for interaction times, we calculated *ego art*, which measures the average response time of the user, and *alter art*, which is the average response time of all other users to user *i* [50].

We also considered two sets of variables to measure the language's complexity (Section 3.2.2) and identify different dimensions of the language content (Section 3.2.3).

**3.2.2. Language complexity and informativeness.** Over the years, numerous studies have used textual metrics to explore the complexity of human language from social media posts [51–53]. In line with these works, we have measured user language complexity based on the three variables. The first is *Word Count*, representing the total number of words users post. It is a proxy for language richness and complexity [54, 55]. Building upon the findings of Lewis & Frank [51] and Owens & Wedeking [52], we included the variable *Six-Letters*. This variable measures the percentage of words in each post with a length greater than six letters. A higher share of long words is associated with communicative sophistication and thus increases language complexity [52]. Lewis & Frank [51] also supported the inclusion of this variable, which demonstrated that longer words refer to conceptually more complex meanings. The first two variables proved to be connected to the educational background of people. Several authors, such as Béland et al. [56], Le Dorze and al. [57], and Mackenizie [58] shed light on the positive impact of education on the length of text used to describe images or the completeness of their descriptions. Consequently, *Word Count* and *Six-Letters* can be interpreted as proxies for educational attainment. Higher linguistic complexity monitored with these variables can also indicate higher user education [59]. Lastly, we introduced a measure of the average informativeness of a user's post, i.e., *Document Informativeness*. It is calculated using the term frequency-inverse document frequency (TF-IDF) metric. The idea is that user posts contribute novel information when they incorporate words not commonly found in other posts and when the intended message is effectively conveyed without uninformative text. Therefore, the frequency of occurrence of each word is multiplied by the logarithmically scaled inverse fraction of the posts that contain that word. The informativeness for each post analyzed is calculated as follows:

$$Document\ Informativeness = \frac{1}{n}\sum_{w \in C} f_w \log \frac{N}{n_w}$$

Where $N$ represents the total number of documents in the corpus (referring to a user's posts); $n$ is the total number of words that appear on a user's post; $C$ indicates the set of these words; $f_w$ is the frequency of a specific word $w$, and $n_w$ is the number of posts where the word $w$ appears.

We employed standard text preprocessing techniques before calculating the three metrics (word count, six letters, and document informativeness). This involved eliminating stop-words, which typically add little value to the meaning of a sentence, as well as punctuation and special characters. Additionally, we converted all words to lowercase and extracted stems by removing word affixes using the Natural Language Toolkit (NLTK) Snowball Stemmer algorithm [60], as recommended by Jivani [61].

**3.2.3. Sentiment and psychological aspects of natural language use.** The language individuals use daily can reveal significant aspects of their social and psychological spheres [62]. To uncover psychological markers from users' posts on Reddit, we use a quantitative approach based on the idea that language characteristics can be counted and statistically analyzed [63–

66]. Our quantitative text analysis is based on previous evidence that words people use convey psychological information or represent their emotional states or opinions. Following this approach, we measured language sentiment and other dimensions using the Python programming language and Linguistic Inquiry and Word Count (LIWC) software [67].

The selected variables fall into two distinct categories of language *style* and *content*–each possessing unique psychometric and psychological properties.

According to the definition of Tausczik and Pennebaker [55], *style words*, also known as function words, consist of auxiliary verbs, interrogative verbs, prepositions, pronouns, and articles, which we explain in the following.

The *auxiliary verbs*, such as "can", "could", "must", "should", "may", "might" and "would", are mainly used in English to express ability, permission, possibility, obligation, necessity, intention, prediction, or probability [68]. By analyzing the frequency of these verbs, we can gain insight into the narrative style of a user's language. Brandt and Herzberg [35] highlighted that a higher occurrence of auxiliary verbs indicates a more dynamic use of language, often involving personal anecdotes.

Other markers of a dynamic narrative style are *pronouns*. We measured the frequency of the first-person pronoun *I*, commonly used when expressing oneself or referring to one's perspective or actions. Its frequency can exhibit associations with factors such as depression, illness, and, more broadly, self-oriented focus [69]. In particular, Bucci & Freedman [70], Rude et al. [71], and Stirman & Pennebaker [72] demonstrated that people who are more vulnerable to depression tend to use first-person pronouns more frequently when expressing their feelings compared to those who are less susceptible to depression. Furthermore, how individuals use pronouns can vary depending on their level of social connection [69, 73]. Using more first-person singular pronouns indicates a more egocentric narrative focus and a more personal communication style. On the other hand, using more second- and third-person pronouns, such as *You* and *She / He*, can represent a user's level of social engagement [62]. This is why we also measured the frequency of second- and third-person singular pronouns.

The use of *Articles*, such as "a", "an", and "the" is another variable that can provide valuable information about the level of detail in the comments. Indeed, it is a proxy of a user's categorical language, which indicates that they use more formal language and provide precise and complex descriptions [35]. Greater use of articles is also associated with users' gender [74], with studies showing that men tend to use more articles than women [75–77].

The dimension *Interrogives* refers to the presence of interrogative words or phrases in comments, such as "who", "what", "where", "when", "why" and "how". This dimension indicates a user's level of curiosity or interest [67].

In addition to these *style words*, the second category of variables consists of *content words* comprising nouns and many descriptive adjectives. From a psychological standpoint, the first category, *style words*, reflects how users communicate, while *content words* convey their expressed opinions (e.g., sentiment), emotional states (e.g., anger, affection, feel) or interests (e.g., leisure) [55]. In the following, we describe the variables belonging to this category.

*Sentiment* is a variable derived from sentiment analysis, a natural language processing technique that aims to extract, convert, and interpret opinions from a text, classifying them as positive, negative, or neutral [78]. We used the VADER lexicon, which is a pre-built sentiment analysis tool that assigns sentiment scores to the entire post. Our measure ranges from -1 (negative) to +1 (positive) and is a crucial aspect in understanding the overall tone of a text.

*Anger* counts the frequency of words that express feelings of anger and aggression, such as "angry", "hateful", "annoyed", and "frustrated". This variable has been incorporated in various studies to map the nature of social media communication before, during, and after the Covid-19 pandemic [79, 80]. A higher presence of these terms indicates worse health. On the other

hand, a branch of literature dedicated to repressive coping argued the opposite [81]. It suggested that individuals who avoid using negative words when describing their emotions are at a greater risk of experiencing subsequent health problems [81].

The *Affection* variable is associated with focusing on emotional states and the well-being of an individual; it refers to the overall emotional tone or mood experienced by a user. Braun et al. [82] found that higher values of positive-affect language characterize texts produced by users with higher emotional intelligence.

*Feel* is related to the emotional tone and expression in language. This variable measures the degree to which individuals use the language of sensations. It includes words such as "hard", "cool", and "felt" [28].

*Leisure* reflects users' engagement in leisure activities and their desire to discuss such topics on social media. This dimension includes words such as "game", "fun", "play", and "party", and it has been utilized in numerous psycholinguistic studies about the Covid-19 pandemic, e.g., Gandino et al. [83] and Su et al. [84].

The presence of words indicative of basic psychological needs, desires, and motivations is measured by *Drives*. This dimension includes words related to achievement, power, affiliation, and other fundamental drives that shape human behavior [67].

*Reward* is related to expressing rewards, incentives, positive goals, and approaches [67]. The idea is that individuals with high-reward attention may be more motivated during the pandemic to take preventive measures to achieve positive health outcomes. This variable has the potential to offer valuable information on the approach taken by various groups of subjects.

*Risk* refers to the extent to which language use reflects a willingness to take chances or engage in risky behavior; it captures the presence of words and phrases associated with risk-taking, adventure, and daring activities [67]. We introduce this dimension to understand how many individuals from the three groups were more risk-oriented before the pandemic.

*Family* is associated with a focus on family dynamics and support; it includes words related to family relationships and roles, such as "mother", "father", "sister", and "brother". Past studies have shown that individuals who use family-related words in their writings are perceived as extraverted [85, 86]. Moreover, valuable research for our work comes from Gutanku et al. [26]. They explored the language of psychological stress with a dataset of social media users and found that stressed users post about family time less frequently than users who are not stressed.

*Differentiation* concerns the amount of words that reflect cognitive complexity and differentiation in language. It includes words such as "or", "but", "if", and "not"[67].

*Insight* represents the extent to which a person gains new understanding, knowledge, or self-awareness about themselves or their experiences; it is related to a user's level of self-reflection and introspection. As suggested by Ogden and Cornwell [54], insight expressions may reflect the participants' willingness to analyze topics that are personal to them.

Lastly, *Health* includes words related to physical and mental health, such as "doctor", "sick", "pain", and "therapy"; it focuses on a user's perception of their physical well-being. Brown et al. [29], Ferguson et al. [30], and Pauli & Alpers [31] demonstrated that individuals with hypochondriacal beliefs tend to process health-related information more extensively than those without such beliefs. Using this dimension, we want to investigate whether individuals who have experienced Long Covid exhibited more significant health concerns before the pandemic; this could imply greater control and thus a greater likelihood of discovering that the preexisting symptoms of Covid-19 are associated with Long Covid.

### 3.3. Overview of study hypotheses and models

To sum up, we used the variables described in Section 3.2.1 to verify the last two hypotheses (*H4-H5*), while the variables included in Sections 3.2.2 and 3.2.3, which are related to the communication style and the psychological aspects of Reddit users, are used to check the first three (*H1-H3*).

To understand the user communication profiles within the No Covid, Covid, and Long Covid groups, we performed the Kruskall-Wallis one-way analysis of variance. Furthermore, pairwise comparisons with a Bonferroni correction were performed.

In addition, we performed logistic regression models to identify the network and textual characteristics that distinguish individuals who have experienced Long Covid from those who have only contracted Covid without developing Long Covid symptoms. Consequently, our dependent variable describes the user's Long Covid class membership. It takes the value of 1 if labelled Long Covid or 0 if the user belongs to the Covid class.

Table 1 provides an overview of the research hypotheses

## 4. Results

In Fig 3 we show the distribution of pre-pandemic posts extracted and analyzed for each group. In total, we collected the posts of 6.107 Reddit Users, of which 2.986 belonged to the Long Covid group, 592 belonged to the Covid group, and the remaining 2.529 belonged to the no Covid group. Consequently, we extracted 984.625 posts, 23% belonging to the Covid group, 32% to the Long covid group, and the remaining 45% to the no covid group. On average, the pre-pandemic posts of each Covid user is 363,09 with a standard deviation of 489,87. Users in the Long Covid group have an average posts count of 104,76 with a standard deviation of 228,34. Lastly, individuals in the No Covid group show an average Reddit post of 166,36 posts, with a standard deviation of 349,87.

As presented in Table 2, our findings indicate that all centrality metrics significantly differ between the three groups. However, the interaction variables (ego art and alter art) were not statistically significant in distinguishing users. Additionally, of the three variables related to language complexity, only six-letter was statistically insignificant. Similarly, the variables you, drives, and reward did not appear to differ among the three groups of users. Individuals who have contracted COVID-19 exhibit elevated values in all variables in social network analysis. Specifically, their communication style is characterized by more auxiliary verbs and articles and a higher frequency of third-person singular pronouns. On the contrary, subjects in the Long Covid group tend to use more first-person singular pronouns and interrogative forms. Regarding content, COVID users discuss family and risk more frequently than their counterparts. In addition, their posts contain more anger-related words, distinguishing them from the other groups. Interestingly, those who develop Long Covid exhibit the lowest average for the

**Table 1. Research hypotheses and variables.**

| Hypothesis | Formulation |
|---|---|
| H1 | Long COVID users express more negative emotions compared to COVID-19 users. |
| H2 | Long COVID users express more health-related topics on social media compared to COVID-19 users. |
| H3 | Long COVID users use a more verbose and direct language compared to COVID-19 users. |
| H4 | Long COVID users have lower social media communication (posting, commenting) activity compared to COVID-19 users. |
| H5 | Long COVID users have lower connection to social media user networks compared to COVID-19 users. |

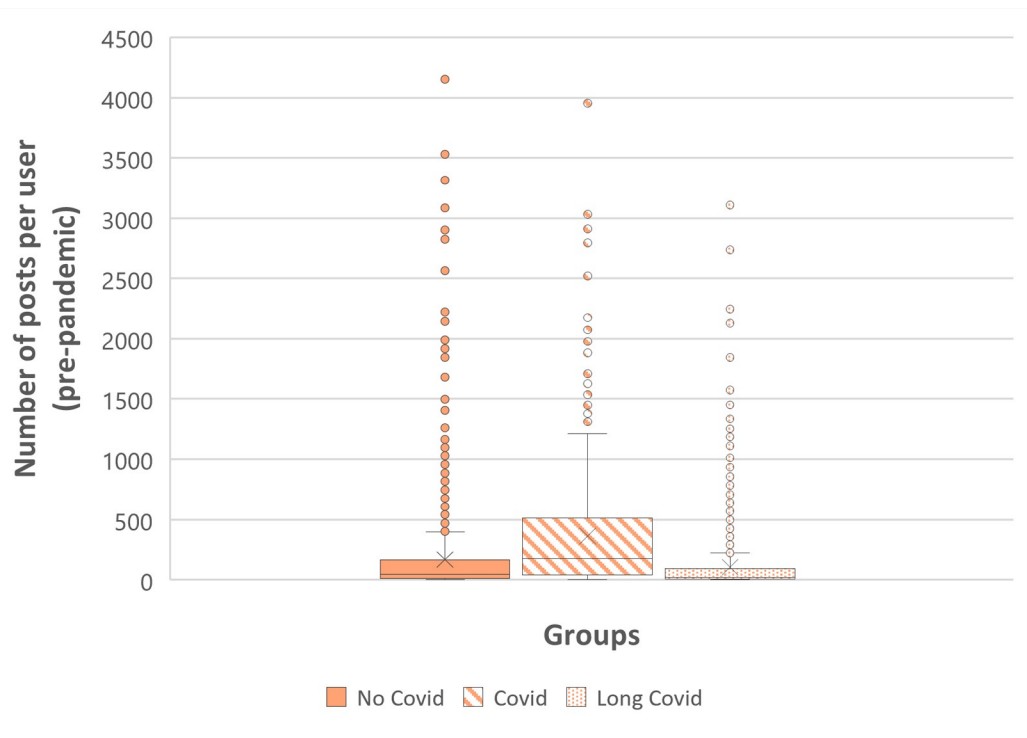

**Fig 3. Number of pre-pandemic posts per user.**

anger variable but the highest for sentiment and feeling. Meanwhile, individuals without COVID have the highest mean for the affected variable; they tend to discuss leisure more frequently but use fewer insight words. However, users in the Long Covid group exhibited a greater focus on health even before the pandemic, as seen from the mean values of this dimension.

We performed seven logistic regression models to test the hypotheses described in Section 2. Table 3 shows an overview of the regression results.

The first three hypotheses aim to explore variations in the psychological aspects of natural language between Long Covid and Covid users. For this purpose, we created Model 1 to test whether there are differences between the use of negative expressions *(H1)*, the use of health-related words *(H2)*, and the use of more verbose and direct language *(H3)*. Consequently, this first model includes all the variables presented in Section 3.2.3. On the other hand, Model 2 examines the impact of language complexity variables to address the third hypothesis. The findings from Model 2 suggest that language complexity variables do not play a significant role in classifying subjects with Long Covid, except for the Word Count. Indeed, the value of McFadden's R2 for this model is the second lowest among the seven models we have created. Looking at the goodness of fit of Model 1, we note that the model explains about 8.1% of the variance in the data. This indicates that the psychological dimensions explain a low percentage of the total variance, leaving a significant proportion unexplained.

The following three models were employed to investigate the impact of Reddit users' social network structures as further social predictors predicting the likelihood of being in the Long Covid group using SNA variables *(H4-H5)*. We constructed three distinct models due to the collinearity of the variables related to the number of messages sent and received by a user, its degree, and betweenness centrality metrics. The results reveal that Model 3 explains more of

**Table 2.** Kruskal-Wallis independent samples tests.

| Variable | Groups | N | Mean | Sig. | Post hoc analysis | | |
|---|---|---|---|---|---|---|---|
| | | | | | No Covid | Covid | Long Covid |
| **Messages sent** | No covid | 2529 | 164.98 | 0.000 | | *** | *** |
| | Covid | 592 | 362.47 | | *** | | *** |
| | Long covid | 2986 | 103.88 | | *** | *** | |
| **Messages received** | No covid | 2529 | 10.22 | 0.000 | | *** | *** |
| | Covid | 592 | 15.47 | | *** | | *** |
| | Long covid | 2986 | 2.55 | | *** | *** | |
| **In degree** | No covid | 2529 | 1.26 | 0.000 | | *** | *** |
| | Covid | 592 | 2.35 | | *** | | *** |
| | Long covid | 2986 | 0.72 | | *** | *** | |
| **Out degree** | No covid | 2529 | 134.04 | 0.000 | | *** | *** |
| | Covid | 592 | 294.92 | | *** | | *** |
| | Long covid | 2986 | 91.28 | | *** | *** | |
| **Betweenness** | No covid | 2529 | 232397.45 | 0.000 | | *** | *** |
| | Covid | 592 | 537101.29 | | *** | | *** |
| | Long covid | 2986 | 222191.16 | | *** | *** | |
| **Ego art** | No covid | 2529 | 0.05 | 0.729 | | | |
| | Covid | 592 | 0.11 | | | | |
| | Long covid | 2986 | 0.01 | | | | |
| **Alter art** | No covid | 2529 | 0.05 | 0.025 | | * | |
| | Covid | 592 | 0.24 | | * | | ** |
| | Long covid | 2986 | 0.02 | | | ** | |
| **Word Count** | No covid | 2529 | 21.63 | 0.000 | | *** | *** |
| | Covid | 592 | 29.17 | | *** | | *** |
| | Long covid | 2986 | 34.75 | | *** | *** | |
| **Six Letter** | No covid | 2529 | 0.27 | 0.904 | | *** | *** |
| | Covid | 592 | 0.26 | | *** | | |
| | Long covid | 2986 | 0.26 | | *** | | |
| **Document Informativeness** | No covid | 2529 | 5.13 | 0.000 | | *** | *** |
| | Covid | 592 | 4.94 | | *** | | |
| | Long covid | 2986 | 4.89 | | *** | | |
| **Auxiliary Verbs** | No covid | 2529 | 7.370 | 0.000 | | *** | *** |
| | Covid | 592 | 8.383 | | *** | | *** |
| | Long covid | 2986 | 8.044 | | *** | *** | |
| **I** | No covid | 2529 | 4.294 | 0.000 | | *** | *** |
| | Covid | 592 | 3.754 | | *** | | *** |
| | Long covid | 2986 | 4.644 | | *** | *** | |
| **You** | No covid | 2529 | 2.470 | 0.066 | | | |
| | Covid | 592 | 2.302 | | | | |
| | Long covid | 2986 | 2.492 | | | | |
| **She/He** | No covid | 2529 | 0.922 | 0.000 | | *** | ** |
| | Covid | 592 | 1.062 | | *** | | *** |
| | Long covid | 2986 | 0.986 | | ** | *** | |
| **Articles** | No covid | 2529 | 4.840 | 0.000 | | *** | *** |
| | Covid | 592 | 5.704 | | *** | | *** |
| | Long covid | 2986 | 5.327 | | *** | *** | |

*(Continued)*

**Table 2.** (Continued)

| Variable | Groups | N | Mean | Sig. | Post hoc analysis | | |
|---|---|---|---|---|---|---|---|
| | | | | | No Covid | Covid | Long Covid |
| **Interrogatives** | No covid | 2529 | 1.583 | 0.000 | | *** | ** |
| | Covid | 592 | 1.623 | | *** | | * |
| | Long covid | 2986 | 1.646 | | ** | * | |
| **Sentiment** | No covid | 2529 | 0.144 | 0.000 | | *** | *** |
| | Covid | 592 | 0.117 | | *** | | *** |
| | Long covid | 2986 | 0.184 | | *** | *** | |
| **Anger** | No covid | 2529 | 1.08 | 0.000 | | *** | *** |
| | Covid | 592 | 1.10 | | *** | | *** |
| | Long covid | 2986 | 0.84 | | *** | *** | |
| **Affection** | No covid | 2529 | 8.95 | 0.000 | | *** | *** |
| | Covid | 592 | 7.57 | | *** | | *** |
| | Long covid | 2986 | 8.36 | | *** | *** | |
| **Feel** | No covid | 2529 | 0.69 | 0.000 | | | *** |
| | Covid | 592 | 0.55 | | | | *** |
| | Long covid | 2986 | 0.71 | | *** | *** | |
| **Leisure** | No covid | 2529 | 1.96 | 0.000 | | * | *** |
| | Covid | 592 | 1.74 | | * | | *** |
| | Long covid | 2986 | 1.48 | | *** | *** | |
| **Drives** | No covid | 2529 | 7.304 | 0.133 | | | |
| | Covid | 592 | 7.018 | | | | |
| | Long covid | 2986 | 7.076 | | | | |
| **Reward** | No covid | 2529 | 1.883 | 0.872 | | | |
| | Covid | 592 | 1.732 | | | | |
| | Long covid | 2986 | 1.831 | | | | |
| **Risk** | No covid | 2529 | 0.473 | 0.000 | | *** | *** |
| | Covid | 592 | 0.595 | | *** | | *** |
| | Long covid | 2986 | 0.546 | | *** | *** | |
| **Family** | No covid | 2529 | 0.41 | 0.000 | | *** | * |
| | Covid | 592 | 0.45 | | *** | | *** |
| | Long covid | 2986 | 0.39 | | * | *** | |
| **Differentiation** | No covid | 2529 | 2.650 | 0.000 | | *** | *** |
| | Covid | 592 | 3.204 | | *** | | *** |
| | Long covid | 2986 | 3.052 | | *** | *** | |
| **Insight** | No covid | 2529 | 1.921 | 0.000 | | *** | *** |
| | Covid | 592 | 2.038 | | *** | | *** |
| | Long covid | 2986 | 2.268 | | *** | *** | |
| **Health** | No covid | 2529 | 0.49 | 0.000 | | *** | *** |
| | Covid | 592 | 0.59 | | *** | | *** |
| | Long covid | 2986 | 0.88 | | *** | *** | |

Note: Independent-Samples Kruskal-Wallis Test. Asymptotic significances are displayed. Post hoc analysis: pairwise comparisons with a Bonferroni correction.

*** $p < 0.001$

** $p < 0.01$

* $p < 0.05$

**Table 3. Logistic regression models.**

| Variable | Model 1 | Model 2 | Model 3 | Model 4 | Model 5 | Model 6 | Model 7 |
|---|---|---|---|---|---|---|---|
| | Odds ratio (95% CI) | Odds ratio (95% CI) | Odds ratio (95% CI) | Odds ratio (95% CI) | Odds ratio (95% CI) | Odds ratio (95% CI) | Odds ratio (95% CI) |
| Auxiliary Verbs | 0.927*** (0.888–0.967) | | | | | 0.924** (0.881–0.970) | 0.926*** (0.887–0.966) |
| I | 1.149*** (1.093–1.208) | | | | | 1.111*** (1.049–1.175) | 1.123*** (1.070–1.191) |
| You | 1.012 (0.948–1.080) | | | | | 1.002 (0.933–1.076) | |
| She/He | 1.034 (0.951–1.124) | | | | | 1.056 (0.960–1.161) | |
| Articles | 0.937* (0.891–0.986) | | | | | 0.922** (0.871–0.977) | 0.942* (0.895–0.992) |
| Interrog | 1.086 (0.996–1.184) | | | | | 1.112* (1.009–1.226) | 1.111* (1.003–1.230) |
| Sentiment | 6.970*** (3.439–14.127) | | | | | 3.596** (1.547–8.360) | 3.850** (1.716–8.64) |
| Anger | 0.847** (0.767–0.937) | | | | | 0.873* (0.782–0.974) | 0.889 (0.799–0.988) |
| Affection | 1.041* (1.004–1.079) | | | | | 1.065** (1.021–1.111) | 1.073** (1.031–1.117) |
| Feel | 1.228* (1.027–1.469) | | | | | 1.306* (1.053–1.619) | 1.324* (1.065–1.647) |
| Leisure | 0.936** (0.890–0.984) | | | | | 0.954 (0.906–1.006) | |
| Drives | 1.036 (0.989–1.085) | | | | | 1.0345 (0.982–1.089) | |
| Reward | 0.933 (0.854–1.018) | | | | | 0.944 (0.854–1.042) | |
| Risk | 0.938 (0.834–1.048) | | | | | 0.937 (0.831–1.057) | |
| Family | 0.819** (0.717–0.935) | | | | | 0.811* (0.706–0.933) | 0.830** (0.733–0.940) |
| Differtiation | 0.921* (0.856–0.990) | | | | | 0.889** (0.820–0.964) | |
| Insight | 1.120* (1.014–1.237) | | | | | 1.082 (0.969–1.208) | |
| Health | 1.486*** (1.269–1.740) | | | | | 1.517*** (0.970–1.208) | 1.542*** (1.291–1.841) |
| Word Count | | 1.012*** (1.008–1.017) | | | | 1.012*** (1.006–1.018) | 1.010*** (1.005–1.016) |
| Six Letter | | 2.222 (0.426–11.682) | | | | 5.277 (1.006–1.018) | 4.020 (0.578–27.952) |
| Document Informativeness | | 0.851* (0.745–0.972) | | | | 0.884 (0.726–1.075) | |
| Messages sent | | | 0.998*** (0.998–0.998) | | | 0.998*** (0.998–0.999) | 0.998*** (0.998–0.999) |
| Messages received | | | 0.989*** (0.982–0.995) | | | 0.986*** (0.980–0.993) | 0.987*** (0.981–0.993) |
| In degree | | | | 0.870*** | | | |
| Out degree | | | | 0.998*** | | | |
| Betweenness | | | | | 1* (0.999–1) | | |
| Ego art | | | 1.016 (0.958–1.077) | 1.012 (0.956–1.072) | 0.956 (0.897–1.019) | 1.001 (0.933–1.074) | |

*(Continued)*

**Table 3.** (Continued)

| Variable | Model 1 | Model 2 | Model 3 | Model 4 | Model 5 | Model 6 | Model 7 |
|---|---|---|---|---|---|---|---|
| | Odds ratio (95% CI) | Odds ratio (95% CI) | Odds ratio (95% CI) | Odds ratio (95% CI) | Odds ratio (95% CI) | Odds ratio (95% CI) | Odds ratio (95% CI) |
| Alter art | | | 0.982 (0.933–1.034) | 0.975 (0.932–1.020) | 0.948 (0.905–0.992) | 0.987 (0.937–1.040) | |
| Constant | 2.946*** (1.566–5.539) | 6.149*** (3.254–11.621) | 7.855*** (7.026–8.783) | 8.331*** (7.407–9.370) | 5.132*** (4.695–5.611) | 4.407 (0.841–23.083) | 1.752 (0.728–4.218) |
| McFadden's R2 | 0.081 | 0.018 | 0.093 | 0.0892 | 0.004 | 0.160 | 0.155 |
| N | 3578 | 3578 | 3578 | 3578 | 3578 | 3578 | 3578 |

Note: Odds ratios from logistic regression analysis to determine the impact of our variables on the probability of belonging to the Long Covid group. CI: Confidence Interval.

\*\*\* $p < 0.001$

\*\* $p < 0.01$

\* $p < 0.05$

the analyzed phenomenon, which justifies our choice of considering the messages sent and messages received instead of in degree and out degree in the final models. Furthermore, we found that the average response time is not significant in making a distinction between the Covid and Long Covid groups, unlike betweenness centrality.

To capture the maximum amount of information and comprehensively understand the phenomenon, we included all the variables considered for analysis in the sixth model. It is the full model, whereas Model 7 is more parsimonious and only includes the best combination of significant variables. McFadden's R2 is significantly higher for these two models than for those considering only the separate block of variables, thus suggesting that the study of social interaction and the analysis of linguistic characteristics play a role in distinguishing people in the Covid and Long Covid groups.

## 5. Discussion

From the evidence of the Kruskal-Wallis analysis, we can deduce that users who stated that they developed symptoms of COVID-19 have the highest level of messaging activity among the three groups. This group interacts with more users within their social network, suggesting a broader range of connections and engagement. We also note that these users have the highest values of linguistic complexity, a higher number of auxiliary verbs, and articles characterizing their posts. This indicates a more formal and precise communicative style [35]. Based on their post content, we note that they tend to utilize more expressions related to anger, risk, and family compared to users who have not had COVID-19 or users with Long Covid. Hence, even before the pandemic, they express or discuss their frustrations, dissatisfaction, or negative emotions more frequently than other groups. This finding aligns with several studies linking negative emotions, such as anger, to an increased likelihood of developing a disease, e.g., Cohen et al. [24], De Choudhury & De [19], and Shen et al. [20]. Moreover, these users used more words associated with risk, probably because they were more aware or concerned about potential life risks. The increased usage of expressions related to the family could reflect their personal values and the significance of familiar relationships in their lives.

Users in the No-COVID group tend to have a more extensive and refined communication vocabulary. Their posts are typically more intricate and informative while also incorporating emotional language. There is a greater emphasis on leisure activities within their contents, while they use fewer words related to the category *insight*. These results are consistent with the

finding of Pennebaker et al. [87], who demonstrated that decreased use of insight words, e.g., "know", "how", and "think", is associated with better health conditions. In addition, individuals who have not developed COVID symptoms are likely to be more engaged in leisure activities, implying higher levels of positive affect and life satisfaction and, consequently, higher life well-being [27].

Consistent with the findings of the regression models, our work suggests that individuals in the Long Covid class generally exhibit lower messaging activity than those with only Covid, both in terms of sending and receiving messages. The signal coming from the betweenness centrality values also agrees with this. The Long Covid users have less messaging activity. On the other hand, individuals in the Covid class play a more critical role in facilitating communication between other network users. This result confirms our hypotheses 4 and 5 regarding the lower involvement of Long Covid users in their social network than those in the Covid class. Moreover, individuals in the Long Covid class tend to express themselves using more words in their posts. Despite the length of their comments, there are fewer articles and auxiliary verbs but many interrogative forms. These linguistic characteristics offer valuable insights into their communication style. For instance, the reduced usage of articles may suggest a direct and concise communication approach, where conveying information takes precedence over specific details. Similarly, the decrease in auxiliary verbs might indicate a preference for straightforward and assertive statements, focusing on essential information rather than hypothetical scenarios [35].

A odds ratio greater than one for the interrogative dimension suggests a higher tendency to ask questions or seek information from others in their online communication [67]. This result confirms our hypothesis 3. Moreover, it is in line with the studies of Suedfeld & Rank [88] about the "disruptive stress hypothesis". They supported the idea that when people experience significant stress, their thinking becomes less complex. According to the disruptive stress hypothesis, individuals with high stress levels might try to solve problems with simpler, less complicated thinking because stress disrupts and simplifies information processing. Consequently, since the communication style of individuals reflects the type of their thoughts, we could associate the simpler communication style with the higher stress level of users who belong to the Long Covid class. The heightened frequency of singular pronouns underscores the significance of personal experiences, thoughts, and emotions shared by these individuals [69]. This evidence is confirmed by the significant dominance of variables like feel and affect.

Long Covid users use more emotional language, compared to users with Covid, which can include a wide range of emotions, such as happiness or excitement, and tend to share in their posts the general state of the feelings they are experiencing. Among the emotions that are not significant for identifying Long Covid subjects is anger. Long Covids' comments are sparse in words such as "angry", "hate", "annoyed", and "frustrated". At the same time, the overall sentiment of their posts tends to be predominantly positive. The lower frequency of negative emotions in the posts of users with Long Covid compared to those with Covid denies our hypothesis 1, but it is consistent with the literature on repressive coping [81]. According to this theory, people who do not use words with negative emotions are at a higher risk of subsequent health problems than those who use at least some words with negative emotions. Moreover, it is interesting to consider these results in light of past research that states that individuals more vulnerable to depression tend to use first-person pronouns more frequently when expressing their feelings compared to individuals who are less susceptible to depression [70–72]. This could indicate that individuals who develop Long Covid are already more likely to experience intense emotional distress before the onset of the pandemic. In addition, how individuals utilize pronouns can vary depending on the level of social connection [69, 73]. As previously mentioned, users of Long Covid tend to employ more first-person singular pronouns,

indicating a more self-centered narrative focus and a more personal communication style. An odds ratio lower than one of the family measure also shows that discussing family relationships on social media is not a prominent characteristic or topic for individuals in the Long Covid class. Based on the result of Gutanku et al. [26], which associated the lower occurrence of family-related words with more significant stress in individuals, it is plausible to suggest that Long Covid users are more susceptible to stress. In contrast, in the content posted before the pandemic, subjects in the Long Covid class were already mainly talking about health. This finding implies that health was already a significant concern or interest for these individuals even before their experience with Long Covid. It suggests that individuals who develop Long Covid symptoms may have a greater tendency towards hypochondria. Research conducted by Brown et al. [29], Ferguson et al. [30], and Pauli & Alpers [31] have shown that individuals with hypochondriacal beliefs tend to process health-related information more extensively than those without such beliefs. This heightened processing may facilitate the retrieval and written elaboration of health-related information. It is also aligned with Zhao & Zhou [25], who reported that users posting and searching for solely Covid-related content demonstrated worse mental health. Moreover, the preexisting focus on health-related concepts may have empowered them with a greater sense of control, leading to their eventual discovery of having Long Covid symptoms. Our analysis shows that Long Covid users express health-related topics more than Covid users. This confirms the validity of our hypothesis 2.

Fig 4 provides an overview of the hypotheses formulated and examined in this study, differentiating between those that were validated and those that were not supported by our findings.

## 6. Conclusions

Through this research, we have expanded our understanding of some possible social determinants of Long Covid, contributing valuable insights for developing effective prevention strategies.

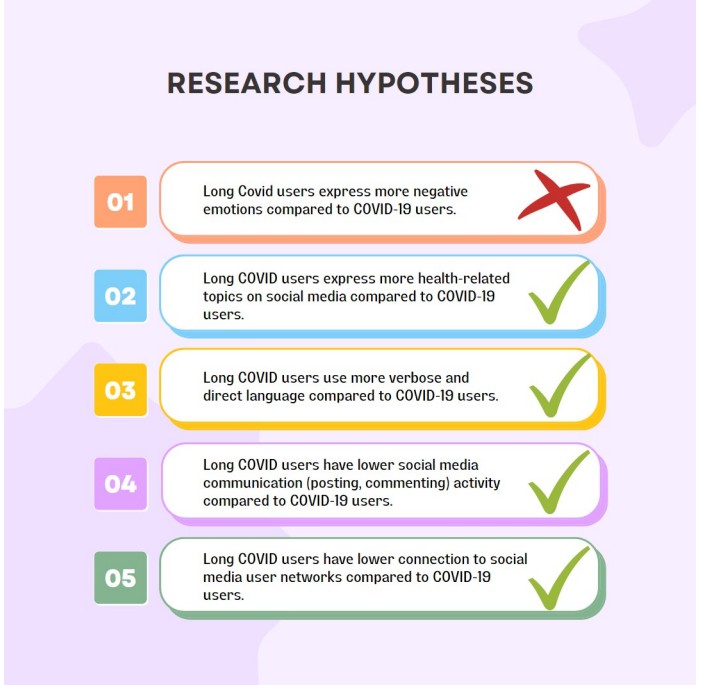

**Fig 4. Supported and unsupported hypotheses.**

## 6.1. Theoretical and practical implications

By analyzing social media data from Reddit user profiles, we offer unique perspectives on potential risk factors associated with Long Covid. Since our investigation delves into the shifts in communication styles and network structures from Covid to Long Covid users, it has broader implications for Covid studies. Indeed, it reveals that individuals developing COVID-19 exhibit a wider range of connections and engagement within their social network, elevated levels of linguistic complexity, and a higher tendency to use expressions related to negative emotions. Moreover, our study helps to create psychological profiles related to Long Covid patients, highlighting that people who developed Long Covid often discussed health-related topics before the pandemic. Hence, it suggests that those users may have a greater tendency towards hypochondria, which aligns with previous research on how hypochondriacal belief impacts disease progression [29, 30].

Our analysis leverages an automated methodology incorporating social network analysis, text mining, and natural language processing. The data we selected offers a significant advantage in characterizing the communication and psychological profile of individuals who have reported contracting COVID-19 and developing Long Covid, providing a perspective directly centred on patients' voices.

In particular, the findings discussed in the previous Section can be integrated into patient history records, capturing data produced when patients were unaware of being observed. By extracting characteristics from social media, we can reduce the bias inherent in traditional survey instruments like structured questionnaires, leading to more informed decisions. As Zhao and Zhou [25] suggest, if these investigation methodologies are designed and applied on a large scale, it can enhance preventive and reactive intervention measures in health crisis management. Furthermore, the profiling of the three user groups appears to align with existing theories regarding patient lifestyle management. This information may offer healthcare providers additional data to support their decisions and medical advice for patients. For instance, the findings from the No-Covid group suggest that engaging in physical activity might help prevent infectious diseases like COVID-19. Encouraging an active lifestyle that includes various recreational activities could be beneficial. These proactive measures may enhance immune function and reduce the risk of experiencing long-term effects after a COVID-19 infection [89].

Finally, in light of the health crisis caused by COVID-19, our research provides evidence to support the effectiveness of using analytical tools based on social media data. Indeed, an automated tool that allows healthcare providers to quickly extract patient information and categorize them could greatly help prioritize interventions at such critical times, optimizing the patient management process.

## 6.2. Limitations and future perspectives

This work has some limitations that also point to future research direction.Exploring different social media platforms with diverse demographics could help reduce any biases resulting from Reddit's potential misrepresentation of the general population. For example, Reddit does not allow us to extract information about users' geo-location. In addition, due to the lack of descriptive information about the users, we could not gather additional data on factors such as age, gender, or level of education, which might affect, for example, the probability of having Long Covid symptoms. Introducing these control variables would enhance the study's robustness. In addition, our choices at the user classification step can be enhanced by incorporating additional criteria. Future work could involve expanding the selection to include subreddits dedicated to Long COVID, thereby capturing active users in other forums focused on related discussions. To enhance the robustness of our dictionary for detecting user-declared COVID-

19 infections, future work could involve expanding keyword variants, incorporating idiomatic expressions, updating symptom-related terms, and integrating machine learning models for contextual understanding.

Additionally, our reliance on the Reddit dataset limits our ability to include information about pre-existing health conditions that may increase susceptibility to developing Long Covid. As Jacobs et al. [90] suggested, it is crucial to consider comorbid conditions such as asthma, chronic constipation, reflux, seasonal allergies, rheumatoid arthritis, depression/anxiety, in addition to age, gender, race, and smoking. These factors proved to be significantly associated with the development of Long Covid. Therefore, future research should explore incorporating additional variables beyond those proposed in our current study. One potential approach is to complement our social media analytics with interviews and questionnaires targeting individuals who have experienced COVID-19 symptoms and Long Covid. An ideal strategy would involve mapping their ego networks while comprehensively analyzing the discrepancies between online social networks and real-life social networks. Another interesting analysis might also consider different categories of people with Long Covid, based on their symptoms. It could be that our predictors may more accurately anticipate psychological aspects related to the disease, such as depression or feeling tired, rather than more physiological aspects, such as a taste or hearing disorder.

[89], Finally, although this study focused on factors predicting the likelihood of Long Covid, those factors may also be appropriate to predict general health anxiety, especially considering the heterogeneity of Long Covid symptoms [91]. As illustrated in the introduction, individuals affected by Long Covid often exhibit symptoms similar to those of depression or migraine, including headache and anxiety, and thus, similar predictors [3, 4, 12]. This overlap could suggest that the same factors predicting Long Covid might also influence broader health diseases. Therefore, future research should aim to provide more detailed insights into the specific symptoms and predictors differentiating Long Covid from other illnesses, employing an approach that prioritizes depth over breadth. This, in turn, could lead to more accurate treatment and recovery strategies and could further inform target interventions.

## Supporting information

**S1 Fig. Examples of the regular expressions used to detect user COVID-19 infection.** S1 Fig provides examples of regular expressions employed to identify mentions of COVID-19 infection within user posts. Regular expressions (regex) are patterns used to match character combinations in strings. In the context of detecting COVID-19 infection mentions, these regex patterns are designed to capture a variety of ways users might refer to their infection status. (TIF)

**S2 Fig. Example of COVID-19 infection indicators.** S2 Fig illustrates in blue an example of indicators we used to identify mentions of COVID-19 infection in user posts. (TIF)

## Author Contributions

**Conceptualization:** Peter A. Gloor.

**Data curation:** Ludovica Segneri, Nandor Babina, Andrea Fronzetti Colladon.

**Formal analysis:** Ludovica Segneri, Nandor Babina, Teresa Hammerschmidt, Andrea Fronzetti Colladon.

**Investigation:** Ludovica Segneri, Nandor Babina.

**Methodology:** Teresa Hammerschmidt, Andrea Fronzetti Colladon, Peter A. Gloor.

**Project administration:** Peter A. Gloor.

**Software:** Nandor Babina.

**Supervision:** Andrea Fronzetti Colladon, Peter A. Gloor.

**Validation:** Ludovica Segneri.

**Writing – original draft:** Ludovica Segneri, Andrea Fronzetti Colladon.

**Writing – review & editing:** Ludovica Segneri, Teresa Hammerschmidt, Andrea Fronzetti Colladon, Peter A. Gloor.

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
