## [Decision Letter · Decision Letter 0]

6 Feb 2024

PONE-D-23-37009Too much focus on your health might be bad for your health! – Reddit user’s communication style predicts their Long COVID likelihoodPLOS ONE

Dear Dr. Gloor,

Thank you for submitting your manuscript to PLOS ONE. After careful consideration, we feel that it has merit but does not fully meet PLOS ONE’s publication criteria as it currently stands. Therefore, we invite you to submit a revised version of the manuscript that addresses the points raised during the review process.

**The manuscript has been evaluated by two reviewers, and their comments are available below**.

**The reviewers have raised a number of major concerns, mostly relating to methodological and analytic aspects of your submission**.

**Could you please carefully revise the manuscript to address all comments raised?**

We look forward to receiving your revised manuscript.

Kind regards,

Steve Zimmerman, PhD

Senior Editor, PLOS ONE

Journal Requirements:

2. In the online submission form you indicate that your data is not available for proprietary reasons and have provided a contact point for accessing this data. Please note that your current contact point is a co-author on this manuscript. According to our Data Policy, the contact point must not be an author on the manuscript and must be an institutional contact, ideally not an individual. Please revise your data statement to a non-author institutional point of contact, such as a data access or ethics committee, and send this to us via return email. Please also include contact information for the third party organization, and please include the full citation of where the data can be found.

Reviewers' comments:

Reviewer's Responses to Questions

**Comments to the Author**

1. Is the manuscript technically sound, and do the data support the conclusions?

Reviewer #1: Partly

Reviewer #2: Partly

2. Has the statistical analysis been performed appropriately and rigorously? 

Reviewer #1: Yes

Reviewer #2: No

3. Have the authors made all data underlying the findings in their manuscript fully available?

Reviewer #1: Yes

Reviewer #2: No

4. Is the manuscript presented in an intelligible fashion and written in standard English?

Reviewer #1: Yes

Reviewer #2: Yes

5. Review Comments to the Author

Reviewer #1: The authors conducted an interesting study to explore social predictors influencing the potential for long COVID-19 among Reddit users. They used text and Social Network Analysis on 6,107 Reddit user profiles and comments. The testing of 5 hypotheses adds a structured approach to their analyses. A few suggestions for the authors' consideration to enhance the clarity and impact of their work.

1. The study categorized the users into those who had Long COVID, those never contracted covid, and those who claimed to have covid without developing long covid. To strengthen the reader's confidence in these classifications, it would be beneficial to detail how they verified the accuracy of these groups. For instance, how did they ensure that individuals in the COVID group did not exhibit long covid symptoms? Was there a validation process using a secondary dataset? Clarifying these aspects would substantiate their findings and methodologies.

2 While the paper is rich in data and analysis, the organization of the results could be enhanced for better reader comprehension. Aligning the presentation of results with the respective hypotheses they test would streamline the narrative and highlight your key findings more effectively. This re-organization could prevent important insights from being overshadowed by the extensive information provided.

Reviewer #2: Review of the article PONE-D-23-37009

"Too much focus on your health might be bad for your health! – Reddit user's communication style predicts their Long COVID likelihood"

This study is interesting. The theoretical background is well-explained and comprehensive. Unfortunately, there is a lack of reporting on the methods and the results section. Moreover, crucial confounding factors are not discussed in the study's limitations.

Methods

There is no description of the study design.

Settings: from which part of the world were the Reddit posts?

Page 14. In the methodology section, How was the random selection process performed?

How many people were identified as members of a Covid-long and Covid forum? How many of them were followed up?

Page16. About language complexity. Nothing is mentioned about language complexity being highly influenced by education.

I did not find any mention of inclusion criteria. Which were they? There is mention of English auxiliary verbs. Does it mean this study includes only English posts?

Page 18. The use of more formal language. Again, the use of more precise and formal language can be strongly determined by education.

Page 20, table 1. H2 Why is the dependent variable here COVID-19 and not Long Covid? The hypothesis H2 concerns Long COVID.

H3 Why is No Covid the control variable?

Page 22, It is likely there is bias. Maybe the long COVID group suffered more frequently from comorbidity. See this publication. https://www.ncbi.nlm.nih.gov/pmc/articles/PMC9816074/

How many posts were extracted by participant? All of them before pandemic?

Results

Page 24 Table 2. It says "Session 2". It should say "Section 2."

The models are not explained. The results are explained neither. Describe all values in the footnote or the title. For example, the values are coefficients. There are no confidence intervals of these coefficients. Why is it reported coefficients (I guess so, looking at the values) and not Odds ratios (+- 95% confidence intervals?)? The tables should be explained so they are understood independently from the manuscript text.

Page 25. What were the statistical criteria for choosing the "best" model? McFadden's R2 is higher in model 6 than in model 7.

Figure 1. It is misleading to name the control group a "Randomized control group". This name is used for experimental studies.

Page 26.

Discussion

"Table 4 provides an overview of the differences in the communication style distinguishing Long Covid users from COVID and No-COVID users." This sentence should be moved to the results section.

Page 29 discussion. The section "limitations" lacks a lot of reporting. Nothing is discussed about essential confounders, such as education, socioeconomic status, and comorbidity.

Nothing is commented on selection bias, which is very likely in these studies. No efforts have been made to deal with bias.

6. PLOS authors have the option to publish the peer review history of their article (what does this mean?). If published, this will include your full peer review and any attached files.

Reviewer #1: No

Reviewer #2: **Yes: **Gloria A Aguayo

---

## [Author Response · Author response to Decision Letter 0]

20 Feb 2024

We have done our best to follow the PLOS ONE style requirements:

• Insertion of continuous line numbering.

• Proper formatting of paragraph headings.

• Converted references to Vancouver style as suggested.

• References to figures were changed (Fig 1 instead of Figure 1).

• Changed the position of table captions.

• Converted figures to TIFF format.

As this was an anonymous analysis with publicly available and aggregate data and, thus, no way of identifying individuals. Hence, it was not deemed necessary to obtain IRB approval. Therefore, there is no ethics statement from an IRB. Nevertheless, HSLU ( University of Applied Sciences Lucerne) approved the Master of Science thesis proposal of one of the authors, which was the basis of the data analyzed in this paper. 

The dataset containing the numerical computations underlying the statistical models presented in the paper is now available at: 10.6084/m9.figshare.25251316

---

## [Decision Letter · Decision Letter 1]

13 May 2024

PONE-D-23-37009R1Too much focus on your health might be bad for your health! – Reddit user’s communication style predicts their Long COVID likelihoodPLOS ONE

Dear Dr. Gloor,

Thank you for submitting your manuscript to PLOS ONE. After careful consideration, we feel that it has merit but does not fully meet PLOS ONE’s publication criteria as it currently stands. Therefore, we invite you to submit a revised version of the manuscript that addresses the points raised during the review process.

We look forward to receiving your revised manuscript.

Kind regards,

Michal Ptaszynski, PhD

Academic Editor

PLOS ONE

Journal Requirements:

Reviewers' comments:

Reviewer's Responses to Questions

**Comments to the Author**

1. If the authors have adequately addressed your comments raised in a previous round of review and you feel that this manuscript is now acceptable for publication, you may indicate that here to bypass the “Comments to the Author” section, enter your conflict of interest statement in the “Confidential to Editor” section, and submit your "Accept" recommendation.

Reviewer #1: All comments have been addressed

Reviewer #3: (No Response)

2. Is the manuscript technically sound, and do the data support the conclusions?

Reviewer #1: Yes

Reviewer #3: Yes

3. Has the statistical analysis been performed appropriately and rigorously? 

Reviewer #1: Yes

Reviewer #3: Yes

4. Have the authors made all data underlying the findings in their manuscript fully available?

Reviewer #1: Yes

Reviewer #3: No

5. Is the manuscript presented in an intelligible fashion and written in standard English?

Reviewer #1: Yes

Reviewer #3: Yes

6. Review Comments to the Author

Reviewer #1: The authors made an effort to address previous comments and contributed to the advancement of the manuscript. The study provides insights into the classification of Reddit users into distinct COVID-19 groups through text analysis and Social Network Analysis. Additionally, the exploration into the potential psychological profiles of these groups based on social media interactions presents an innovative approach to understanding public health dynamics in digital spaces. To further refine the manuscript and maximize its contribution to the field, please considering the following suggestions:

1. Clarification of Public Health and Clinical Implications: The feasibility and effectiveness of the proposed approach in classifying users into different COVID-19 groups have been well demonstrated. The subsequent analysis that reveals diverse psychological profiles among these groups is equally intriguing. However, the broader implications of these findings, particularly in the context of public health and clinical settings, remain somewhat unclear. It would greatly benefit readers to have a clearer exposition of how these insights could be operationalized to inform public health strategies, clinical interventions, or policy-making. Elaborating on these aspects would not only enhance the practical relevance of the work but also underline its significance within the broader discourse on managing health crises through social media analytics.

2. Methodological Transparency and Participant Selection Bias: The key findings regarding the differential engagement and linguistic patterns among the identified COVID-19 groups are compelling. However, the methodology underlying participant identification and group classification warrants further elaboration. Specifically, a more detailed description of the criteria and processes used to allocate users to specific COVID-19 groups would be valuable. This includes any validation mechanisms employed to ensure the accuracy of group assignments. Additionally, considering a significant portion of the study population appears to have been sourced from specific COVID Reddit forums (i.e., 'covidlonghauler' and forums dedicated to COVID-19 positive patients), there is a potential risk of selection bias influencing the results. Providing a breakdown of the participant distribution across these forums and discussing the steps taken to mitigate any resultant bias would strengthen the credibility of the study findings.

Reviewer #3: This paper explores a novel approach to predicting Long COVID, a post-viral condition following COVID-19 infection. The authors investigate whether a user's communication style on the social media platform Reddit can be used to identify individuals at higher risk.

The idea of using communication style to predict a health outcome is innovative and potentially valuable. Utilizing Reddit data allows for access to a large and diverse population, potentially leading to more generalizable findings.

The study needs to address whether communication style is a cause or effect of Long COVID. Individuals experiencing Long COVID symptoms might naturally gravitate towards health-focused online communities.

The paper should clarify how well the communication style markers predict Long COVID compared to other factors: Are these markers specific to Long COVID, or do they indicate general health anxiety?

The findings would benefit from external validation using a different population or data source to assess generalizability.

This paper presents an intriguing concept with the potential to improve early identification of Long COVID. However, further research is needed to address the limitations mentioned above and strengthen the causal link between communication style and Long COVID risk.

The paper would be strengthened by including detailed information on sample size and demographic characteristics of the Reddit user population studied.

It would be also helpful to know what specific aspects of communication style were analyzed.

More importantly, the authors should discuss potential ethical considerations of using social media data for health predictions.

7. PLOS authors have the option to publish the peer review history of their article (what does this mean?). If published, this will include your full peer review and any attached files.

Reviewer #1: No

Reviewer #3: No

---

## [Author Response · Author response to Decision Letter 1]

21 Jun 2024

all concerns have been addressed

---

## [Decision Letter · Decision Letter 2]

23 Jul 2024

Too much focus on your health might be bad for your health! – Reddit user’s communication style predicts their Long COVID likelihood

PONE-D-23-37009R2

Dear Dr. Gloor,

We’re pleased to inform you that your manuscript has been judged scientifically suitable for publication and will be formally accepted for publication once it meets all outstanding technical requirements.

Kind regards,

Michal Ptaszynski, PhD

Academic Editor

PLOS ONE

Additional Editor Comments (optional):

Reviewers' comments:

Reviewer's Responses to Questions

**Comments to the Author**

1. If the authors have adequately addressed your comments raised in a previous round of review and you feel that this manuscript is now acceptable for publication, you may indicate that here to bypass the “Comments to the Author” section, enter your conflict of interest statement in the “Confidential to Editor” section, and submit your "Accept" recommendation.

Reviewer #1: All comments have been addressed

Reviewer #3: All comments have been addressed

2. Is the manuscript technically sound, and do the data support the conclusions?

Reviewer #1: Yes

Reviewer #3: Yes

3. Has the statistical analysis been performed appropriately and rigorously? 

Reviewer #1: N/A

Reviewer #3: Yes

4. Have the authors made all data underlying the findings in their manuscript fully available?

Reviewer #1: Yes

Reviewer #3: Yes

5. Is the manuscript presented in an intelligible fashion and written in standard English?

Reviewer #1: Yes

Reviewer #3: Yes

6. Review Comments to the Author

Reviewer #1: The authors have addressed the reviewer's main concerns in this revision. However, the clean version of their submitted manuscript has track changes.

Reviewer #3: (No Response)

7. PLOS authors have the option to publish the peer review history of their article (what does this mean?). If published, this will include your full peer review and any attached files.

Reviewer #1: No

Reviewer #3: No

---

## [Editor Report · Acceptance letter]

25 Jul 2024

PONE-D-23-37009R2 

PLOS ONE

Dear Dr. Gloor, 

I'm pleased to inform you that your manuscript has been deemed suitable for publication in PLOS ONE. Congratulations! Your manuscript is now being handed over to our production team.

Kind regards, 

on behalf of

Dr. Michal Ptaszynski 

Academic Editor

PLOS ONE